# Effect of Genistein on Starch Digestion In Vitro and Its Mechanism of Action

**DOI:** 10.3390/foods13172809

**Published:** 2024-09-04

**Authors:** Jianhui Jia, Boxin Dou, Man Gao, Chujia Zhang, Ying Liu, Na Zhang

**Affiliations:** College of Food Engineering, Harbin University of Commerce, Harbin 150028, China; 2019088@mdjnu.edu.cn (J.J.); dbx0803@163.com (B.D.); gm205055@163.com (M.G.); zhangchujia330@163.com (C.Z.); zhangna@hrbcu.edu.cn (N.Z.)

**Keywords:** genistein, starch digestion, α-amylase, α-glucosidase, inhibition mechanism

## Abstract

The digestive properties of starch are crucial in determining postprandial glycaemic excursions. Genistein, an active phytoestrogen, has the potential to influence starch digestion rates. We investigated the way genistein affected the digestive properties of starch in vitro. We performed enzyme kinetics, fluorescence spectroscopy, molecular docking, and molecular dynamics (MD) simulations for analysing the inhibitory properties of genistein on starch digestive enzymes as well as clarifying relevant mechanism of action. Our findings demonstrated that, following the addition of 10% genistein, the contents of slowly digestible and resistant starches increased by 30.34% and 7.18%, respectively. Genistein inhibited α-amylase and α-glucosidase, with half maximal inhibitory concentrations of 0.69 ± 0.06 and 0.11 ± 0.04 mg/mL, respectively. Genistein exhibits a reversible and non-competitive inhibiting effect on α-amylase, while its inhibition on α-glucosidase is a reversible mixed manner type. Fluorescence spectroscopy indicated that the presence of genistein caused declining fluorescence intensity of the two digestive enzymes. Molecular docking and MD simulations showed that genistein binds spontaneously to α-amylase via hydrogen bonds, hydrophobic interactions, and π-stacking, whereas it binds with α-glucosidase via hydrogen bonds and hydrophobic interactions. These findings suggest the potential for developing genistein as a pharmacologic agent for regulating glycaemic excursions.

## 1. Introduction

The rising incidence of metabolic syndromes such as hyperglycaemia and hyperlipidaemia is closely linked to improvements in the standard of living [1]. Metabolic syndromes are critical drivers of several chronic diseases, such as type 2 diabetes and hypertension [2]. Managing the digestion rate of ingested carbohydrates can be an effective strategy for controlling glycaemic excursions [3]. Starch is the major source of daily energy intake in most countries, taking up almost 40–80% of the human diet [4]. However, starchy foods can lead to higher postprandial glycaemic responses compared to other types of foods [5]. Controlling starch digestion and glycaemic responses through various methods is crucial in designing healthy starch-based foods.

Enzymes that digest starch in our body mainly include α-amylase and α-glucosidase. Inhibiting starch digestive enzymes can effectively slow down starch hydrolysis, which helps to manage blood glucose levels and reduce the prevalence of metabolic disorders [6]. Many compounds exhibit hypoglycaemic bioactivity by inhibiting starch digestive enzymes, and these inhibitors can play a crucial role in preventing diabetic complications [7,8]. Common drugs that act as inhibitors include acarbose [9], voglibose [10], and miglitol [11]. Their mechanism of action is to reduce postprandial blood glucose spikes and fluctuations by delaying the absorption of carbohydrates in the small intestine, and the inhibitors do not cause hypoglycaemia. However, many inhibitors are prone to adverse side effects, such as hypoglycaemia, gastrointestinal disturbances, and hepatotoxicity [12]. Thus, identifying safe, effective, and economical dietary anti-diabetic compounds is very important.

Food-based polyphenols are a class of natural inhibitors of starch digestive enzymes. Polyphenolic compounds have gained widespread popularity owing to their wide bioavailability and anti-hyperglycaemic effects, with few or no side effects [13]. Genistein is predominantly found in fruits, vegetables, and cereals, with its main source in the human diet being soybeans and soya products [14]. In 31 fruits and nuts where genistein was detected, 8 foods contained more than 100 μg/kg wet weight, and 23 contained less than this amount [15]. The concentration of genistein in 51 cereals commonly eaten ranged from 16 to 5788 μg/kg wet weight. The soy flours were rich sources, containing 889 to 1080 mg/kg wet weight [16]. Genistein has several biological activities, including anti-diabetic, anticancer, antioxidant, and osteoporosis prevention [17]. It has a good overall safety profile and has demonstrated a high efficacy profile, especially in clinical trials. In the study by Tousen et al. [18], maternal ingestion of genistein may not negatively affect dams, foetuses, infants, or offspring in the process of growth. Ullmann et al. [19] evaluated the safety, tolerance, and pharmacokinetics of genistein in healthy volunteers in a clinical pilot study and confirmed its safety and good tolerance within the dose range studied. Demir et al. [20] determined the in vitro inhibitory action of certain phenolic compounds on α-amylase and α-glucosidase and confirmed the better inhibitory action of genistein. Li et al. [21] isolated and identified 12 compounds from *Radix puerariae thomsonii*-bound phenolics, of which genistein exhibited the highest oxygen radical absorbance capacity activity and inhibited the two digestive enzymes.

Till date, studies have seldom analysed the effect of genistein on starch digestion and its mechanisms of action. Our study paid attention to such an effect as well as its inhibitory effect on starch hydrolysing enzymes using in vitro experiments. We performed enzyme kinetics for evaluating the inhibitory effect of genistein on starch digestion and adopted fluorescence spectroscopy combined with molecular docking to evaluate the molecular interactions between genistein and starch digestive enzymes. These findings are anticipated to provide a theoretical basis for a more profound comprehension of the inhibitory mechanisms of genistein on starch digestion and the advancement of glucose-lowering functional foods.

## 2. Materials and Methods

### 2.1. Materials

Genistein (≥98% purity), acarbose (≥98% purity), p-nitrophenyl-α-d-glucopyranoside (pNPG), and rice starch (S28581, total starch content of 97.23, protein 0.67, and fat 0.26 g/100 g (expressed on a dry basis)) came from Shanghai Yuanye Biotechnology (Shanghai, China). α-Amylase from porcine pancreas (E.C. 3.2.1.1, 10 U/mg), α-glucosidase from *Saccharomyces cerevisiae* (E.C. 3.2.1.20, 50 U/mg), and amyloglucosidase from *Aspergillus niger* (E.C. 3.2.1.3, 70 U/mg) were offered by Sigma-Aldrich (Shanghai, China). The d-glucose assay kit came from Megazyme International (Wicklow, Ireland). All the chemicals and reagents used were of analytical grade.

### 2.2. Determination of Digestion Properties of Genistein–Starch Mixed System

A particular volume of mixed enzyme solution was prepared in which the specific activities of porcine pancreatic α-amylase and amyloglucosidase were 150 U/mL and 15 U/mL, respectively, and kept at 37 °C. Rice starch (100 mg) was mixed thoroughly with 0, 2.5, 5.0, 7.5, and 10.0 mg genistein to obtain a genistein–starch mixed system. This mixed system was added to 25 mL of mixed enzyme solution to receive incubation at 37 °C and 160 r/min in a shaking water bath. Aliquots from the reaction solution were removed at 0, 20, and 120 min, followed by being assayed using a glucose assay kit (Megazyme). Below is the calculation equation of the rapidly digestible starch (RDS), slowly digestible starch (SDS), and resistant starch (RS) contents:(1)$$\mathrm{RDS}(\%)=\frac{(G_{20}-G_{0})\times 0.9}{\mathrm{TS}}\times 100$$
(2)$$\mathrm{SDS}(\%)=\frac{{(G}_{120}-G_{20})\times 0.9}{\mathrm{TS}}\times 100$$
(3)$$\mathrm{RS}(\%)=\frac{\mathrm{TS}-\mathrm{RDS}-\mathrm{SDS}}{\mathrm{TS}}\times 100$$

In the above equations, G_0_ represents the content of free glucose within starch; G_20_ and G_120_ are the quantities of glucose generated by the mixed enzyme solution after incubation for 20 and 120 min, respectively; TS represents the total starch content in the sample, mg; and 0.9 is the glucose conversion coefficient.

### 2.3. Determination of the Inhibition Rate of Starch Digestive Enzymes by Genistein

For α-amylase, the method of Qin et al. [22] was used with slight modifications (the enzymatic reaction time was different). The experimenters added different amounts of genistein to α-amylase solution (60 U/mL) so that the final concentrations of genistein were 0.5, 1.0, 1.5, 2.0, 2.5, and 3.0 mg/mL. A 5 mL aliquot of the above solution was taken to undergo 10 min of incubation at 37 °C, following by the addition of 5 mL of starch solution (20 mg/mL) that was prepared and kept at 37 °C. Following 15 min of incubation at 37 °C, the reaction solution was removed, and a glucose assay kit was employed for assaying the glucose produced after starch hydrolysis through measuring the changes in the absorbance at 510 nm. Acarbose served as a positive control. Acarbose is a glycosidase inhibitor that is used in the treatment of type 2 diabetes with the molecular formula C_25_H_43_NO_18_. Acarbose is a complex oligosaccharide whose core structure, acarviose, is linked to maltose residues via an α-1,4-glycosidic bond [23]. Acarbose has a strong affinity for α-amylase and α-glucosidase in the intestinal tract and can be used as a competitive substrate to bind with them, inhibit the hydrolysis of starch and polysaccharides, and delay glucose production [24]. Polyphenols can inhibit starch digestion by binding to starch digestive enzymes, which, in turn, controls glycaemic fluctuations. Its mechanism of action is similar to that of acarbose, which is why acarbose is often chosen as a positive control when studying the inhibitory effect of polyphenols on starch digestive enzymes. For α-glucosidase, the protocol of Wang et al. [25] was used with slight modifications (the concentration of the α-glucosidase solution was different). Different amounts of genistein were added to the α-glucosidase solution (0.01 U/mL) so that the final concentrations of genistein were 0.1, 0.2, 0.3, 0.4, 0.5, and 0.6 mg/mL. A 2 mL aliquot of the solution was kept at 37 °C for 10 min, then the addition of 2 mL of pNPG solution (25 mmol/L). The reaction lasted 15 min at 37 °C and was terminated by adding 6 mL of sodium carbonate solution (0.2 mol/L). Absorbance changes were measured and recorded at 405 nm. Acarbose served as a positive control. The inhibition rates of genistein and acarbose on starch digestive enzymes were calculated using Equation (4). IC_50_ was calculated using SPSS 25.0 (IBM) nonlinear regression fitting.
(4)$$\mathrm{Inhibition}\;\mathrm{ratio}\;(\%)\;=\;1-\frac{{\mathrm{OD}}_{\mathrm{sample}}-{\mathrm{OD}}_{\mathrm{sample}\;\mathrm{blank}}}{{\mathrm{OD}}_{\mathrm{enzyme}\;\mathrm{solution}}-{\mathrm{OD}}_{\mathrm{buffer}\;\mathrm{solution}}}\times 100$$
where OD_sample_, OD_sample blank_, and OD_enzyme solution_ represent the absorbance of the solution containing both the sample and the enzyme, the sample solution, and the enzyme solution, respectively. OD_buffer solution_ is the absorbance of the buffer solution.

### 2.4. Determination of Reversibility of Enzyme Inhibition

The protocol of Dong et al. [26] was followed with minor modifications (the substrate concentration was different). For α-amylase, 10, 20, 30, 40, and 50 mg of α-amylase were added to 5 mL of phosphate buffer, followed by 0, 10, and 20 mg of genistein. The assay mixture received 10 min of incubation at 37 °C. The mixture underwent 15 min of incubation after being added with starch solution (5 mL, 20 mg/mL), followed by the determination of the absorbance change. For α-glucosidase, 2 mL of enzyme solutions were added with genistein of various concentrations (0.005, 0.01, 0.015, 0.02, and 0.025 U/mL) and amounts (0, 4, and 8 mg) to receive 10 min of culture at 37 °C. The mixture of the above solution with 2 mL of pNPG solution (25.0 mmol/L) underwent 15 min of incubation, followed by the measurement of its absorbance at 405 nm. We plotted the relevance of enzyme concentration to the enzymatic reaction rate to analyse whether genistein inhibited starch digestive enzymes reversibly.

### 2.5. Enzyme Inhibition Kinetics

The method in the study of Jiang et al. [27] was modified slightly in this study (the concentration of the inhibitor was different). For α-amylase, the mixture of different amounts (0, 10, and 20 mg) of genistein with 5 mL of α-amylase solution (60 U/mL) received 10 min of culture at 37 °C. Thereafter, the assay mixture added with 5 mL of 10, 20, 40, 50, and 100 mg/mL of starch solution underwent 15 min of incubation at 37 °C. The assay was performed using a glucose assay kit, together with the identification of the absorbance change at 510 nm. 

For α-glucosidase, the assay mixture of 2 mL of α-glucosidase solution (0.01 U/mL) with various amounts of genistein (0, 4, and 8 mg) underwent 10 min of culture at 37 °C, which was then added with 2 mL of pNPG solution of different concentrations (5, 10, 15, 20, and 25 mmol/L). The assay mix was incubated further for 15 min at 37 °C. The experimenters terminated the reaction by adding 6 mL of sodium carbonate solution (0.2 mol/L). The absorbance change of the solution was monitored colorimetrically at 405 nm. The type of reversible inhibition of the enzyme by genistein was analysed by plotting the inverse of the enzyme reaction rate, 1/*V*, against the inverse of the substrate concentration, 1/[*S*]. The equations below were adopted for calculating *K*_m_, *K*_IC_, and *K*_IN_, which, respectively, represented the Michaelis constant, competitive inhibition constant, and non-competitive inhibition constant, in mg/mL [28].

Lineweaver–Burk:(5)$$\frac{1}{V}=\frac{K_{m}}{V_{\max}}\cdot \frac{1}{\left[S\right]}+\frac{1}{V_{\max}}$$

Non-competitive reversible inhibition:(6)$$V=\frac{V_{\max}\cdot [S]}{\left(K_{m}+\left[S]\right)\cdot (1+\frac{\left[I\right]}{K_{\mathrm{IN}}}\right)}$$

Mixed reversible inhibition:(7)$$V=\frac{V_{\max}[S]}{K_{m}(1+\frac{\left[I\right]}{K_{\mathrm{IC}}})+[S](1+\frac{\left[I\right]}{K_{\mathrm{IN}}})}$$
where [*S*]—substrate concentration (mg/mL); *V*—reaction velocity [mg/(mL·min)] at different [*S*] values; V_max_—maximal velocity [mg/(mL·min)]; and [*I*] represents the inhibitor concentration (mg/mL).

### 2.6. Fluorescence Spectroscopy

For fluorescence spectroscopic studies, we followed the protocol in the study of Yu et al. [29]. Firstly, 10 mL of α-amylase (30 U/mL) and α-glucosidase (1 U/mL) solutions were sufficiently mixed with different amounts of genistein (0, 5, 10, 20, 30, and 40 mg), and after incubation at 25, 31, and 37 °C for a total of 10 min, the fluorescence intensity was measured (measurement conditions: excitation wavelength: 280 nm; emission wavelength: 300–400 nm; and slit width: 5 nm). The background fluorescence signals for genistein and buffer were subtracted from the emission spectra and the measured fluorescence data were corrected. Fluorescence quenching parameters were calculated using the following equations:(8)$$F_{a}=F_{m}\cdot e^{(A_{\mathrm{ex}}+A_{\mathrm{em}})/2}$$
where *F*_m_ denotes the measured fluorescence data; *F*_a_ represents the adjusted fluorescence data; and *A*_ex_ and *A*_em_ denote the absorbance at the excitation and emission wavelengths, respectively.

Stern–Volmer dynamic impact quenching equation:(9)$$\frac{F_{0}}{F}=1+K_{A}\tau_{0}[Q]$$

Static quenching equation:(10)$$\frac{F_{0}}{F}=1+K_{Q}[Q]$$

Dissociation constant double reciprocal equation:(11)$$\frac{1}{F_{0}-F}=\frac{1}{F_{0}}+\frac{K_{D}}{F_{0}[Q]}$$
(12)$$K_{D}=\frac{1}{K_{A}\tau_{0}}$$

In the above equations, *F*_0_ and *F* denotes the fluorescence intensity of the enzyme that is affected and not affected by genistein, respectively; *K*_A_ represents the apparent quenching rate constant expressed [L/(mol·s)]; τ_0_ represents the fluorophore fluorescence lifetime, which is 10^−8^ s; [*Q*] is the genistein concentration (mol/L); *K*_Q_ represents the static quenching constant (L/mol); and *K*_D_ represents the dissociation constant, mol/L.

Binding and thermodynamic analyses were used for elucidating the mechanisms of enzyme inhibition, including the inhibition of fluorophores by quenchers. The binding constants and the number of binding sites for genistein in starch digestive enzymes at different temperatures were calculated according to Equation (13), and the thermodynamic parameters of genistein interactions with starch digestive enzymes were calculated according to Equations (14) and (15) [30].
(13)$$\mathrm{lg}(\frac{F_{0}-F}{F})=\mathrm{lg}K_{C}+n\mathrm{lg}[Q]$$
where *F*_0_, *F*, and [*Q*] are identical to those in Equation (9). *K*_C_ is the binding constant (L/mol), and *n* denotes the average number of binding sites.
(14)$$\ln\frac{K_{Q2}}{K_{Q1}}=\frac{\Delta H}{R}(\frac{1}{T_{1}}-\frac{1}{T_{2}})$$
(15)ΔG=−RTlnK=ΔH−TΔS

In the above two equations, Δ*H* symbolises the enthalpy change (kJ/mol); Δ*G* symbolises the free energy change (kJ/mol); Δ*S* symbolises the entropy change [J/(mol·K)]; R symbolises the gas constant [8.314 J/(mol·K)]; and *K*_Q1_ and *K*_Q2_ represent the static quenching constants (L/mol) at temperatures *T*_1_ and *T*_2_.

### 2.7. Molecular Docking

Molecular docking assisted in analysing the binding conformation and mode of genistein interaction with starch digestive enzymes. We downloaded porcine pancreatic α-amylase (PDB ID: 1PIF) and *S. cerevisiae* α-glucosidase (PDB ID: 3AJ7) crystal structures from the RSCB Protein Data Bank (https://www.rcsb.org/, accessed on 18 May 2024), and irrelevant ligands were removed using PyMOL 2.3 (DeLano Scientific LLC). The 3D structure of genistein (Compound CID: 5280961) was available on the NCBI PubChem database (https://pubchem.ncbi.nlm.nih.gov, accessed on 20 May 2024). Molecular docking of genistein to starch digestive enzymes was performed using AutoDock 4.2 (TSRI, San Diego, CA, USA), with semi-flexible docking after eliminating water molecules and supplementing polar hydrogen, and Lamarckian genetic algorithms were conducted with 100 docking tests. PyMOL 2.3 was used for visualisation, and non-covalent interactions were analysed using the PLIP Web tool 2.3.0 [31].

### 2.8. MD Simulation

AmberTools 20 was invoked on the Yinfu Cloud Computing Platform (https://cloud.yinfotek.com/, accessed on 10 June 2024) to perform MD simulations. The system underwent solvation by the TIP3P water model [32], and Na^+^ was employed for achieving the neutralisation of the net charge. The energy was minimised by 10,000 steps of the steepest descent together with 10,000 steps of the conjugate gradient. A 20 ps canonical ensemble (NVT) simulation was adopted to achieve the gradual heating of the system to 300 K. After that come the 2 steps of the equilibration phases: (1) a 200 ps isobaric–isothermal ensemble (NPT) simulation with heavy atoms constrained, then (2) a 500 ps NVT simulation with no constraints. A Berendsen thermostat maintained the temperature at 300 K (coupling constant: 1 ps) and a Monte Carlo barostat maintained the pressure at 1 atm (relaxation time: 1 ps). At last, a 50 ns NVT simulation was conducted, and the time step was 2 fs. Analysis of the root-mean-square deviation (RMSD) and root-mean-square fluctuation (RMSF) relied on the CPPTRAJ [33] module. Calculation of the binding free energy relied on the MM/PBSA (Molecular Mechanics/Poisson–Boltzmann Surface Area) module [34].

### 2.9. Statistical Analyses

All experiments were repeated three times. Data were presented in the form of mean ± standard deviation. The results of the trial were statistically analysed by virtue of SPSS version 25.0 (IBM Corp., Armonk, NY, USA). *p* < 0.05 reported statistical significance. Origin 2022 took charge of graph plotting (Originlab Corp., Northampton, MA, USA).

## 3. Results and Discussion

### 3.1. Effect of Genistein on Digestive Properties of Starch In Vitro

Examination of the direct effect of genistein on the in vitro starch digestive properties utilizing starch as the enzyme/substrate has rarely been reported. As shown in Figure 1, the in vitro digestive properties of the mixed system were significantly altered by the addition of genistein. Starch had the initial RDS, SDS, and RS contents of 58.15, 39.89, and 1.96%, respectively. As the concentration of genistein increased, the RDS content steadily declined, but the other two contents of SDS showed a gradual elevation (*p* < 0.05). Hence, genistein inhibited starch digestion in a dose-dependent manner. As the concentration of genistein reached 0.4 mg/mL, the RDS, SDS, and RS contents of the mixed system were 20.63, 70.23, and 9.14%, respectively. The addition of genistein reduced the RDS content in starch by 37.52% and increased the SDS and RS contents by 30.34 and 7.18%, respectively. In the study by Subaitha et al. [35], increased polyphenol content of millets significantly improved their inhibitory effect on starch digestive enzymes, resulting in a reduced postprandial glycaemic response. Polyphenols reduce starch digestion by binding to sites specific on the starch digestive enzymes, which may also lead to V-starch formation by altering the semi-crystalline layer of starch, impacting its digestive activities of starch in vitro [36].

### 3.2. Inhibitory Ability of Genistein on Starch Digestive Enzymes

The reduction or loss of enzymatic activity due to changes in the nature of essential groups in the enzyme molecule as a result of the influence of certain substances is called inhibition. The inhibition of starch digestive enzymes by different concentrations of genistein and acarbose (positive controls) is shown in Figure 2. Though its inhibitory effect was not as strong as that of acarbose, genistein nevertheless had a good effect on the two digestive enzymes.

Zheng et al. [37] observed that genistein inhibited α-amylase in fermented grapefruit peels, leading to improved anti-diabetic efficacy. Ewa et al. [38] selected certain amino acids to synthesise a co-amorphous system with genistein, which could more effectively inhibit α-glucosidase relative to acarbose. The inhibitory ability of genistein against the two digestive enzymes was significantly enhanced with increasing concentration (*p* < 0.05), indicating that genistein significantly inhibited the enzyme activities. Genistein exhibited a higher IC_50_ on α-amylase (0.69 ± 0.06 mg/mL) than that on α-glucosidase (0.11 ± 0.04 mg/mL). Zheng et al. [39] found that ferulic acid strongly inhibited α-amylase (IC_50_: 0.622 mg/mL) and α-glucosidase (IC_50_: 0.866 mg/mL). The extract of Artemisia abrotanum had a pronounced α-amylase inhibitory activity (IC_50_: 1881.21 mg/mL) and α-glucosidase inhibitory activity (IC_50_: 1171.16 mg/mL) [40]. Compared to them, genistein has a smaller IC_50_ value and good inhibitory activity against starch digestive enzymes. Genistein inhibits α-glucosidase more effectively than α-amylase, possibly because its conformation matches α-amylase to a lesser extent than α-glucosidase [41]. In addition, acarbose presented a weaker inhibition on α-amylase (IC_50_: 28.12 ± 1.59 μg/mL) than that on α-glucosidase (IC_50_: 5.81 ± 0.29 μg/mL).

### 3.3. Reversibility of Genistein’s Inhibitory Effect on Starch Digestive Enzymes

Enzyme inhibition can be classified as either reversible or irreversible, which can occur through various mechanisms. The reversible mechanism involves non-covalent interactions with the enzyme, allowing it to regain activity, while irreversible inhibition results in strong covalent bonds with the inhibitor, causing permanent inactivation while the inhibitor–enzyme complex remains stable [42]. Plotting enzyme concentration against the rate of the enzymatic reaction determined the inhibition type. As can be seen from Figure 3, the straight lines representing different concentrations of genistein all intersect at the origin, and the slope of the straight lines presents a gradual decline as genistein concentration increases, indicating that genistein suppresses starch digestive enzymes in a reversible manner and that genistein reversibly binds to the two enzymes through non-covalent bonds.

### 3.4. Reversible Inhibition Type of Genistein on Starch Digestive Enzymes

The kinetic curve for the inhibition of starch digestive enzymes by genistein is shown in Figure 4. Reversible inhibition has four types: anti-competitive, non-competitive, competitive, and mixed [43]. For α-amylase (Figure 4A), the Lineweaver–Burk curves for different concentrations of genistein showed good linearity. The three lines intersect at a point in the X-axis negative direction. As the concentration of genistein increased, the slope of the straight line increased, *K*_m_ remained constant, and *V*_max_ decreased, indicating that genistein exhibits inhibitory effect on α-amylase in a non-competitive reversible manner. The inhibition degree depends on the genistein concentration and *K*_IN_ and is independent of starch concentration and *K*_m_; lowering the concentration of the inhibitor reduces inhibition [44]. Thus, genistein apparently binds to essential groups other than the enzyme’s active centre to produce non-competitive inhibition, and there is no competition between the substrate and the inhibitor. The Michaelis constant *K*_m_ indicates the affinity held by the enzyme to the substrate; greater *K*_m_ means lower affinity. The *K*_m_ was 29.77 ± 1.62 mg/mL at inhibitor concentrations of 0, 1, and 2 mg/mL, demonstrating that the affinity of α-amylase for starch remained constant as the concentration of genistein increased. *K*_IN_ represents the binding ability of the inhibitor to the free enzyme or enzyme–substrate complex. The smaller the *K*_IN_, the greater the ability, and the more effective the inhibition [45]. The *K*_IN_ of genistein was 13.66 ± 0.97 mg/mL, which was significantly lower than the *K*_m_ value.

α-glucosidase had a linear Lineweaver–Burk curve (Figure 4B), with the intersecting point in the second quadrant. As the concentration of genistein increased, the straight line slope increased, the *K*_m_ increased, and the *V*_max_ decreased; hence, the inhibition of genistein against α-glucosidase was mixed reversible, and the degree of inhibition was between competitive and non-competitive inhibition [46]. Genistein can either compete for the binding group at the active site of the enzyme active site, producing competitive inhibition, or bind to essential groups outside the active site, producing non-competitive inhibition. The *K*_m_ was 1.51, 2.98, and 4.15 mg/mL at inhibitor concentrations of 0, 1, and 2 mg/mL, respectively, suggesting that the affinity of α-glucosidase for its substrate, pNPG, decreased significantly with increasing genistein concentrations. The inhibition constants, *K*_IC_ and *K*_IN_, can measure the binding ability of the inhibitor with the enzyme, and the *K*_IC_ of genistein was 0.71 ± 0.03 mg/mL, and the *K*_IN_ was 4.94 ± 0.36 mg/mL. The *K*_IC_ values were significantly lower than the *K*_IN_ values, indicating that competitive inhibition dominated the mixed reversible inhibition; that is, the binding of genistein to α-glucosidase was more stable than that to the α-glucosidase–pNPG complex.

### 3.5. Fluorescence Quenching Effect of Genistein on α-Glucosidase

#### 3.5.1. Analysis of Quenching Types

Aromatic amino acids (tyrosine, tryptophan, and phenylalanine) in starch digestive enzymes produce endogenous fluorescence at certain excitation wavelengths. The fluorescence intensity and the maximum emission wavelength position have a strong association with the enzyme folding state and the microenvironment of specific amino acid residues [47]. According to Figure 5, the maximum endogenous fluorescence peaks were observed at the excitation and emission wavelengths of 280 and 342 nm for α-amylase and at the excitation and emission wavelengths of 280 and 336 nm for α-glucosidase. The fluorescence intensity of the two digestive enzymes diminished slowly as genistein concentration increased, showing a certain regularity, and for the mixed system, the maximum emission wavelength was slightly red-shifted. The fluorescence intensity of genistein and the buffer were very low and were subtracted from the emission spectrum. The above results indicate that genistein can quench the endogenous fluorescence of the enzyme by interacting with α-amylase and α-glucosidase, and the slight redshift suggests reduction in hydrophobic regions on the surface of the enzyme molecule due to conformational change induced by interaction with genistein. It is also possible that genistein causes a change in the microenvironment around the chromophores that exposure of the buried hydrophilic amino acid residues, increasing the polarity and decreasing the hydrophobicity [48].

Fluorescence quenching types include dynamic, static, and hybrid dynamic–static quenching. Dynamic quenching occurs due to dynamic collision between the quencher and the excited-state fluorophore, which reduces the fluorescence intensity, whereas the quencher interacts with the ground-state fluorophore to draw forth the static quenching to form a complex that reduces the fluorescence intensity. The type of Stern–Volmer quenching curve, the change of static quenching constants at different temperatures, and the magnitude of apparent quenching rate constants are important for determining the type of quenching [49].

For α-amylase (Figure 6A), the Stern–Volmer curve obtained by plotting *F*_0_/*F* against [*Q*] showed a good linear fit, which suggests that there is a genistein-accessible fluorophore in α-amylase and that the quenching is predominantly of a single quenching type. The linear fitting curves at different temperatures were close to parallel, the static quenching constant decreased slightly with increasing temperature of the reaction system, and the quenching type was in accordance with the typical characteristics of the static quenching type [50]. The apparent quenching rate constants (Table 1) were 5.41 × 10^11^, 5.32 × 10^11^, and 4.84 × 10^11^ L/(mol·s), respectively, remarkably larger than the maximum collisional quenching rate constant for biomolecules, 2 × 10^10^ L/(mol·s), that is to say, genistein exerted a static quenching on α-amylase.

For α-glucosidase (Figure 6B), the test points of the Stern–Volmer equation were slightly curved upwards, the correlation coefficients of the linear fitting curves at different temperatures were all >0.95 and the curves were close to parallel, and the static quenching constants tended to rise slightly, so that the type of quenching was a mixed dynamic–static quenching with predominantly static type. Fluorescence quenching of α-glucosidase by genistein obtained by affinity ultrafiltration in black soybean has been reported to be of the static quenching type [51]; however, such different conclusions may be caused by differences in the purity and concentration of the inhibitors and experimental parameters, such as quenching temperature and the specific activity of the enzyme [52].

Because the quenching of α-glucosidase by genistein is not a perfectly static and test points deviate somewhat from linearity, the *K*_A_ calculated using the static quenching equation is not precise enough. Therefore, the double inverse equation (Equation (11)) was used to obtain the graph in Figure 7. The quenching curve in Figure 7 is linear, and the dissociation constant *K*_D_ derived from it is more consistent with the fluorescence quenching process. The *K*_A_ derived from Equation (12) at 25, 31, and 37 °C, were 0.41 × 10^11^, 0.63 × 10^11^, and 1.73 × 10^11^ L/(mol·s), respectively, remarkably larger than the maximum collisional quenching rate constant for biomolecules of 2 × 10^10^ L/(mol·s). Hence, the quenching of α-glucosidase by genistein is a mixed quenching dominated by the static quenching type.

#### 3.5.2. Binding Force and Thermodynamic Analyses

The binding constant, *K*_C_, indicates the quenching affinity exhibited by the inhibitor to the enzyme, and it is positively correlated with the quenching affinity. In Table 1, the binding constants decrease with an increase in temperature. This suggests that genistein and starch digestive enzymes form unstable complexes that get perturbed with a rise in temperature, making the inhibitor less interact with the enzyme and reducing the complex stability. This is consistent with the general characteristics of both static and mixed quenching types dominated by static quenching [53]. The higher binding constant for α-glucosidase than that for α-amylase indicates that genistein inhibits α-glucosidase strongly but inhibits α-amylase in a slightly weaker manner. The number of binding sites for an inhibitor on an enzyme represents the average number of mutually independent and equivalent binding sites available for binding. The number of binding sites at different temperatures (Table 1) reveals that genistein has a single major binding site on the two enzymes. With increasing temperature, the number of binding sites for α-amylase decreased from 1.19 to 0.77 and the number of binding sites for α-glucosidase decreased from 0.76 to 0.58, indicating that increasing temperature affects the quenching regarding the binding process and reduces the binding stability, which is consistent with the results of the static quenching constants and binding constants.

In Table 1, the Δ*G* for the two digestive enzymes at different temperatures was negative, indicating that the binding of genistein to two digestive enzymes can occur spontaneously. Specific to α-amylase, Δ*H* is less than zero and behaves as an exothermic process, and the quenching process is mainly influenced by the static binding of the molecule. Specific to α-glucosidase, Δ*H* > 0, which behaves as an endothermic process, the quenching process is affected by dynamic molecular collisions. The Δ*S* of two digestive enzymes was > 0 regardless of the temperature, indicating an increase in the disorder of the reaction system.

### 3.6. Docking Conformation of Genistein with Starch Digestive Enzymes

As shown in Figure 8, genistein binds to α-amylase and α-glucosidase, forming hydrophobic pockets by secondary structural modifications that encapsulate genistein and generate a complex [54]. The binding site of α-amylase is constructed from residues Trp-58, Trp-59, Glu-60, Tyr-62, Gln-63, Val-163, Leu-165, Arg-195, Asp-197, Ala-198, Glu-233, His-299, and Asp-300. The binding site for α-glucosidase was constructed from residues Lys-156, Gly-160, Gly-161, Asp-233, Asn-235, Ser-236, Thr-237, Trp-238, Phe-314, Asn-317, Asn-415, Ala-418, Ile-419, Glu-422, His-423, Glu-428, Glu-429, Met-430, Lys-432, and Phe-433. Under the influence of genistein, the microenvironment in which the aromatic amino acid residues of the binding site get altered, which may be important for quenching starch digestive enzymes’ fluorescence intensity by genistein [55].

### 3.7. MD Simulations

As shown in Figure 9, genistein binds to α-amylase and α-glucosidase. After 50 ns of MD simulation, it was observed that the composite system rapidly reached a steady state within a relatively short period of time, and the RMSD value was maintained at an amplitude fluctuation of about 0.1–0.2 nm. Meanwhile, compared with α-amylase in the bound state (Figure 9A), free α-amylase had a lower RMSD value and was more stable. This suggests that genistein binding has a destabilising effect on the conformation of α-amylase, thereby increasing the magnitude of fluctuations in the protein structure. This phenomenon stems from the binding mode and mechanical properties of genistein to α-amylase. When genistein binds to α-amylase, it may introduce external forces or change the enzyme conformation, resulting in structural instability, leading to higher RMSD values, indicating greater structural fluctuations. The RMSD of α-glucosidase bound to genistein (Figure 9B) showed the opposite result, with α-glucosidase in the free state having a higher RMSD suggesting instability. This reflects the specific interaction mechanism between genistein and α-glucosidase: genistein binding may stabilise the conformation of α-glucosidase and minimise its conformational fluctuations. This is consistent with the findings of a previous study where it was suggested that in the presence of genistein, α-glucosidase prefers to maintain a more stable conformation for interaction with genistein [56]. In general, both the overall and partial structures of the composite system reached a steady state after 50 ns of MD simulation. This suggests that the simulated time range was sufficient to capture the process of complex formation and stabilisation.

Comparative results of RMSF values of residues of the two digestive enzymes in bound and unbound states further elucidated the details of the restricted mobility of these molecules. Unbound α-amylase (Figure 10A) and α-glucosidase (Figure 10B) had larger amplitudes, while α-amylase and α-glucosidase in the bound state behaved relatively much steadier. The larger amplitude of the α-amylase structure in the unbound state implies that the enzyme may have more flexible active sites in the absence of bound genistein, which may aid the enzyme’s substrate recognition and catalysis [57]. In the unbound state, α-glucosidase has a broad amplitude and a flexible binding target site with numerous random coils, α-helices, and β-folds, making it more mobile. The larger amplitude of the α-glucosidase structure in the unbound state may facilitate substrate recognition and catalysis of the enzyme, thereby allowing it to adapt to different substrate structures and sizes.

### 3.8. Analysis of the Binding Mode of Genistein to Starch Digestive Enzymes

Electrostatic interactions (ΔE_ele_), van der Waals forces (ΔE_vdw_), and non-polar solvation energy (ΔG_nonpol_) facilitate the binding of the two compounds and thus may help to stabilise the binding between genistein and α-amylase and α-glucosidase. However, polar solvation (ΔG_pol_) was not beneficial for the binding process. According to the MD simulation results, calculation of the free energy of the binding of genistein to α-amylase/α-glucosidase (Table 2) was conducted with the MM/GBSA method.

Figure 11A shows that there are three main interactions between genistein and α-amylase: hydrophobic interactions, π-stacking, and hydrogen bonding (H-bonds). First, hydrophobic interactions are formed between Trp-59 of α-amylase and genistein A ring C6 and C10 with C-C distances of 0.304 and 0.346 nm, respectively; between Tyr-62 and C2′ and C3′ of the genistein B ring; between Trp-58 and C5′ of the genistein B ring; and between Asp-300 and C6′ of the genistein B ring. Second, the A ring of genistein forms an π-stacking with the hydrophobic amino acid Trp-59, with the stacking being offset type. The ring centre spacing was 0.401 nm, the ring face angle was 24.72°, and the ring centre offset was 0.100 nm. There is an effective π–π interaction, which is a weak stacking interaction, and offset stacking contributes to stabilising the bound conformation of the complex [58]. The combination of hydrophobic interactions and π-stacking leads to the contribution of van der Waals forces with an energy change (Table 2) of −23.45 kJ/mol (ΔE_vdw_). In addition, strong H-bonding interactions existed between phenolic hydroxyl groups of genistein and specific residues of α-amylase (H-bond lengths of 0.18–0.36 nm). Genistein contains three phenolic hydroxyl groups at C4′, C5, and C7, and the electron delocalisation of the benzene ring makes it susceptible to ionisation and hence H-bonding with other compounds [59]. The Arg-195, Asp-197, and His-299 residues of α-amylase form four H-bonds with the phenolic hydroxyl group at C4′ of the genistein B ring using the side chain as a H-donor (of which two are formed by Arg-195), and the H-bonds formed by the phenolic hydroxyl group of the genistein B ring make an important contribution to anchoring the genistein molecule. The hydroxyl group at C5 of the A ring of genistein acts as a H-donor to form a H-bond with the backbone of the Trp-59 of α-amylase, and the Gln-63 with the side chain acting as a H-donor to form a H-bond with C5 of the A ring of genistein as well. These H-bonds effectively help to maintain a stable binding between genistein and α-amylase, and the energy contribution is −21.90 kJ/mol (ΔE_ele_). The synergistic effect of these interactions resulted in a tight binding between genistein and α-amylase and a strong binding capacity with a free energy change of −20.97 kJ/mol (ΔG_bind_). These results demonstrate a complex mechanism of interaction between genistein and α-amylase, providing key insights into the deeper understanding of this biomolecular interaction.

Figure 11B shows that there are two main types of interactions between genistein and α-glucosidase: hydrophobic and H-bonding. A strong hydrophobic interaction (ΔE_vdw_ = −34.59 kJ/mol) was formed between genistein and α-glucosidase. Residues Glu-429 and Lys-432 of α-glucosidase form hydrophobic interactions at C8 of the genistein A ring, and residue Ile-419 forms a hydrophobic interaction with the genistein B ring at C2′. In addition, genistein formed five H-bonds (ΔE_ele_ = −20.46 kJ/mol) with α-glucosidase with bond lengths of 0.20–0.38 nm. The hydroxyl group at C4′ on the B ring of genistein acts as a hydrogen donor to form H-bonds with the side chain of the Glu-422 of α-glucosidase as well as the main chain of Gly-161; Asn-235 and Asn-317 of α-glucosidase form H-bonds with the A ring of genistein using the side chain as a H-donor; and His-423 uses the side chain as a H-donor to form a H-bond with the C ring of genistein. Together, these interactions maintain a close contact between genistein and α-glucosidase, resulting in a strong binding capacity (ΔG_bind_ = −19.90 kJ/mol). These results indicated the high stability of genistein and α-glucosidase binding. Relying on these interactions, a stable complex is formed between genistein and α-glucosidase, and these features contribute to a better understanding of this biomolecular interaction.

## 4. Conclusions

Our study elucidated that genistein could inhibit α-amylase and α-glucosidase, resulting in lower starch digestibility. Compared to other amylase-inhibiting substances (e.g., ferulic acid, Artemisia abrotanum extracts, etc.), genistein has a strong inhibitory capacity (IC_50_) and is expected to be a potential hypoglycaemic food. On the one hand, genistein binds to α-amylase via hydrophobic interactions, π-stacking, and H-bonds, and the binding region is outside the active centre of the enzyme; the *K*_IN_ of genistein against α-amylase was 13.66 ± 0.97 mg/mL. On the other hand, genistein binds to α-glucosidase only via hydrophobic interactions and H-bonding. The *K*_IC_ and *K*_IN_ of genistein for α-glucosidase were 0.71 ± 0.03 and 4.94 ± 0.36 mg/mL, respectively. This indicates that genistein can spontaneously bind α-amylase and α-glucosidase to form a complex through the conformational change of the enzyme and play an inhibitory role. In summary, genistein is a potential inhibitor of starch digestive enzymes and can be a dietary supplement to retard starch digestion.

## Figures and Tables

**Figure 1 foods-13-02809-f001:**
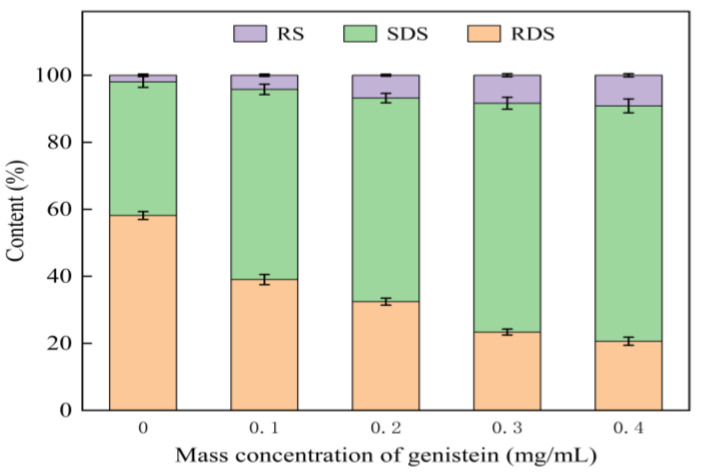
Digestive properties of the genistein–starch mixed system in vitro.

**Figure 2 foods-13-02809-f002:**
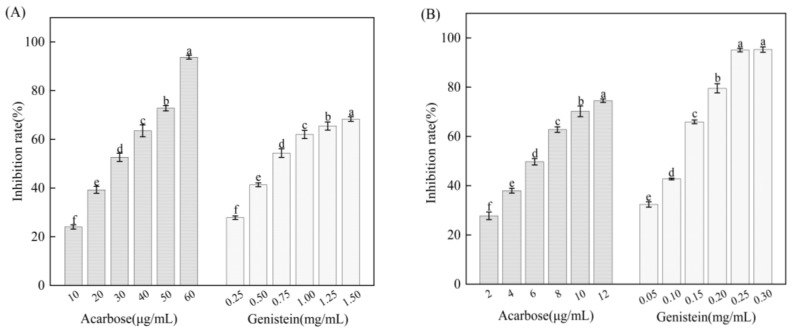
Inhibition rate of α-amylase (**A**) and α-glucosidase (**B**) by genistein and acarbose. Different letters represent a significant difference (*p* < 0.05).

**Figure 3 foods-13-02809-f003:**
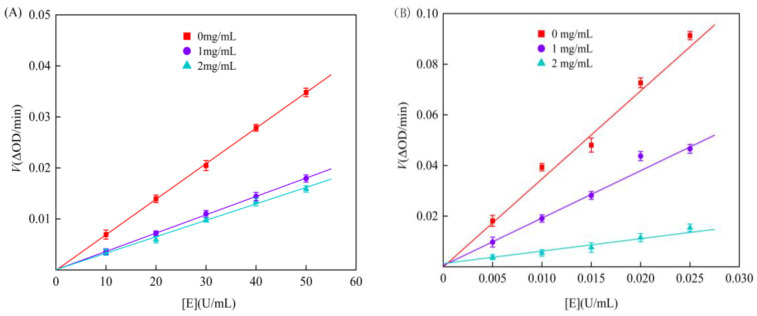
Effect of α-amylase (**A**) and α-glucosidase (**B**) concentrations on reaction rate.

**Figure 4 foods-13-02809-f004:**
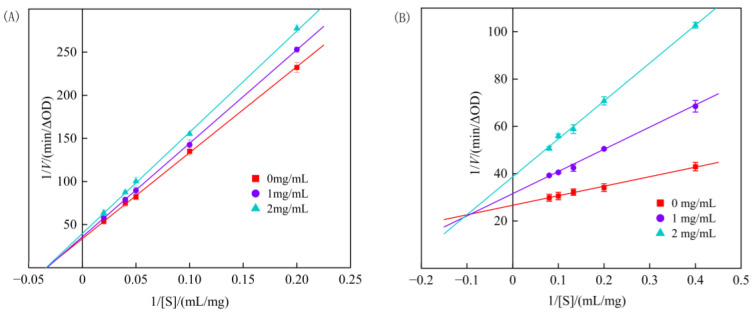
Double reciprocal curve of genistein inhibition of α-amylase (**A**) and α-glucosidase (**B**).

**Figure 5 foods-13-02809-f005:**
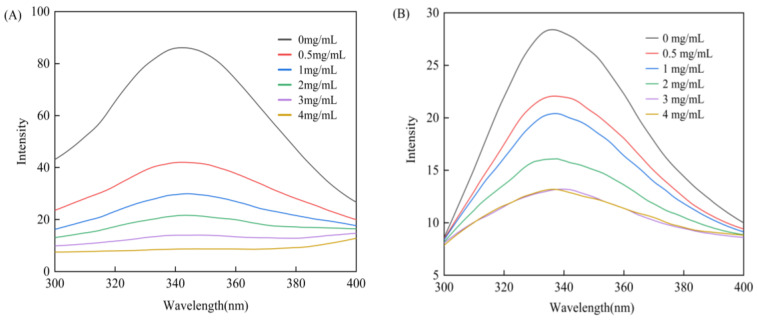
Effect of genistein on the fluorescence spectra of α-amylase (**A**) and α-glucosidase (**B**).

**Figure 6 foods-13-02809-f006:**
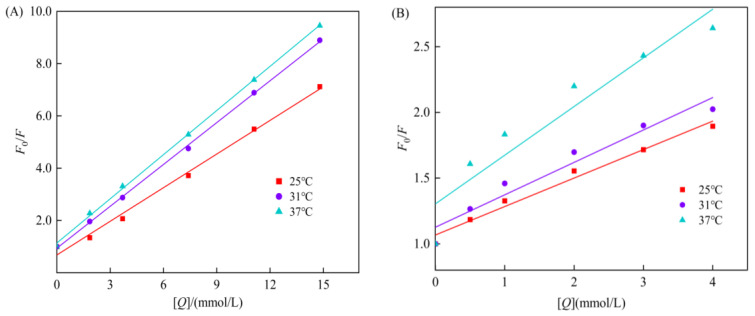
The Stern–Volmer quenching curve of α-amylase (**A**) and α-glucosidase (**B**).

**Figure 7 foods-13-02809-f007:**
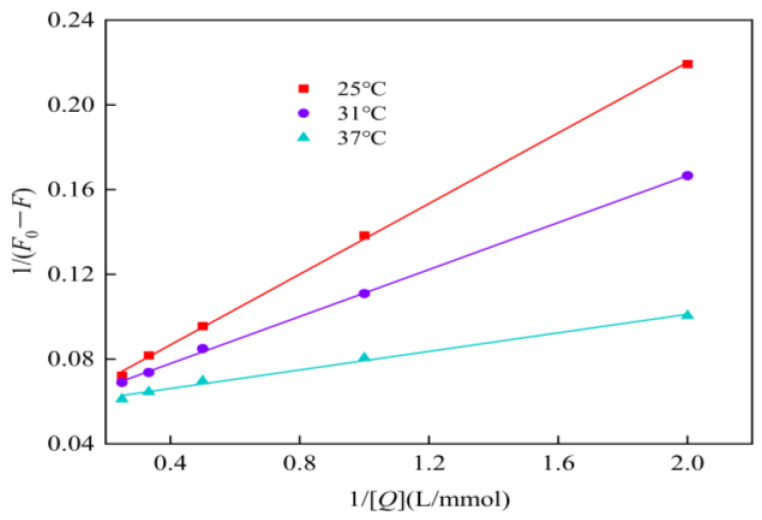
The double reciprocal quenching curve of genistein to α-glucosidase.

**Figure 8 foods-13-02809-f008:**
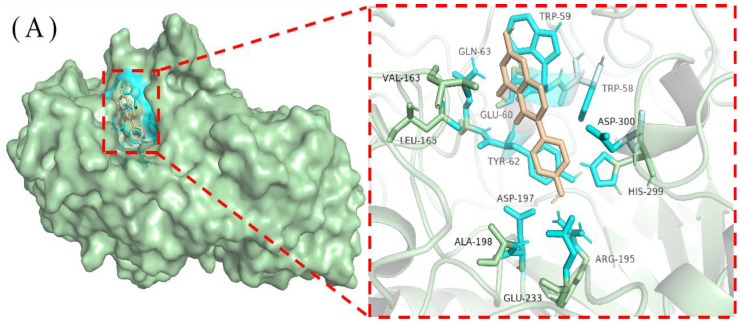
Binding sites and construction residues of α-amylase (**A**) and α-glucosidase (**B**). The enzyme is represented by the pale green surface form; the key residues are represented with cyan sticks; and the genistein is represented with an orange stick.

**Figure 9 foods-13-02809-f009:**
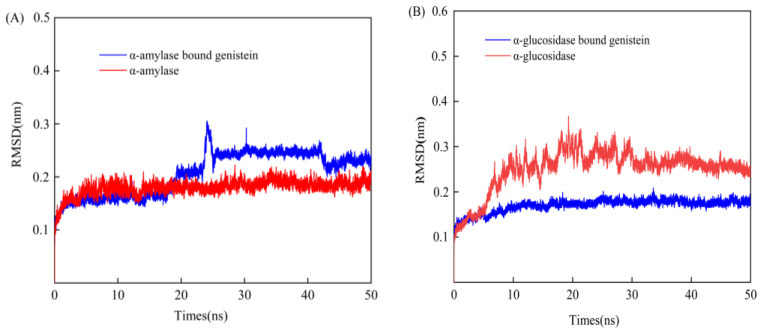
RMSD results of 50 ns MD simulations of genistein with α-amylase (**A**) and α-glucosidase (**B**).

**Figure 10 foods-13-02809-f010:**
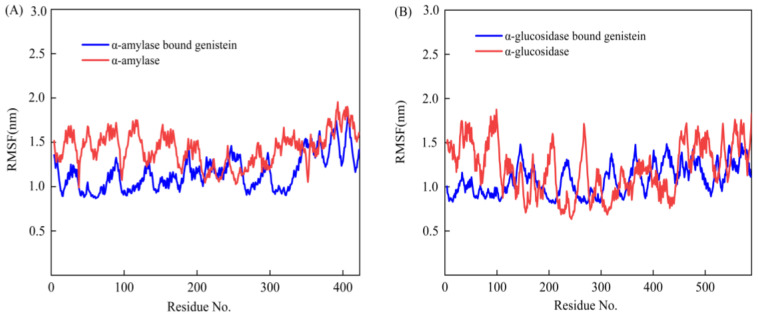
RMSF results of 50 ns MD simulations of genistein with α-amylase (**A**) and α-glucosidase (**B**).

**Figure 11 foods-13-02809-f011:**
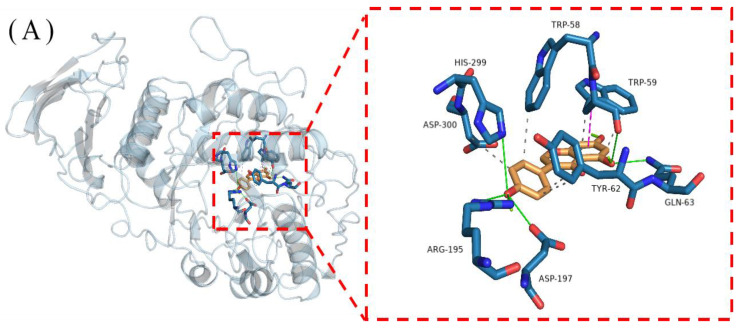
Binding patterns of genistein to α-amylase (**A**) and α-glucosidase (**B**). The hydrophobic interactions are represented with the black dotted line; the hydrogen bonding are represented with the green solid line; and the π-stacking are represented with the pink dotted line.

**Table 1 foods-13-02809-t001:** Fluorescence quenching parameters, binding constants, and thermodynamic parameters.

Enzyme	T (°C)	*K*_Q_ (L·mol^−1^)	*K*_D_ (mol·L^−1^)	*K*_A_ (L·mol^−1^·s^−1^)	*K*_C_ (L·mol^−1^)	*n*	Δ*H* (kJ·mol^−1^)	Δ*S* (J·mol^−1^·K^−1^)	Δ*G* (kJ·mol^−1^)
α-amylase	25	0.54 × 10^3^	0.76 × 10^−3^	5.41 × 10^11^	0.62	1.19	−0.46	50.65	−14.29
	31	0.53 × 10^3^	0.17 × 10^−3^	5.32 × 10^11^	0.46	0.97	−6.02	31.97	−15.55
	37	0.48 × 10^3^	0.11 × 10^−3^	4.84 × 10^11^	0.20	0.77	−2.11	44.59	−15.94
α-glucosidase	25	0.21 × 10^3^	2.44 × 10^−3^	0.41 × 10^11^	61.80	0.76	6.31	67.41	−12.09
	31	0.26 × 10^3^	1.60 × 10^−3^	0.63 × 10^11^	60.71	0.71	8.13	73.50	−13.78
	37	0.29 × 10^3^	0.58 × 10^−3^	1.73 × 10^11^	22.86	0.58	6.85	69.39	−14.66

*K*_Q_: static quenching constant; *K*_D_: dissociation constant; *K*_A_: apparent quenching rate constant; *K*_C_: binding constant; *n*: binding sites; Δ*H*: enthalpy change; Δ*S*: entropy change; Δ*G*: free energy change.

**Table 2 foods-13-02809-t002:** Free energy of binding of the complex (kJ/mol).

Name/Contribution	ΔE_vdw_	ΔE_ele_	ΔG_pol_	ΔG_nonpol_	ΔG_bind_
α-amylase	−23.45	−21.90	27.62	−3.24	−20.97
α-glucosidase	−34.59	−20.46	39.48	−4.33	−19.90

ΔE_vdw_: van der Waals forces; ΔE_ele_: electrostatic interactions; ΔG_pol_: polar solvation energy; ΔG_nonpol_: non-polar solvation energy; and ΔG_bind_: binding energy.

## Data Availability

The data presented in this study are available on request from the corresponding author. The data are not publicly available due to privacy restrictions.

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
