# Peer review of "Effect of Genistein on Starch Digestion In Vitro and Its Mechanism of Action"

_foods, 2024, doi:10.3390/foods13172809_

Round 1

Reviewer 1 Report

Comments and Suggestions for Authors

1. The abstract is too long. 

2. The introduction should be improved. Literature concerning genistein sources and content (ranges of concentrations) must be complemented and presented in detail. Ln 60: "plant-derived - not clear - please elaborate this and support with additional literature.

3. L99-100 purity and composition must be provided. One batch sample was used? Starch must be characterized in detail.

4. Ln 107-122: Is this method described elsewhere? Why such conditions and enzyme activities were applied?

5. Ln 129-132: The method is not referenced. Do you mean the glucose oxidase method? What is the influence of genistein on glucose oxidase???

6. Why exactly acarbose was selected as a positive control?

7. The discussion should be extended - inhibition of amylases by other similar compounds (acarbose, other) should be presented, compared to the genistein. The result of the comparison to other amylase-inhibiting substances should be presented in conclusions to evaluate the potential of genistein.

8. list of references is double numbered.

Reviewer 2 Report

Comments and Suggestions for Authors

All detailed comments are noted in the manuscripts' pdf file
